# ProtoBrainMaps: Prototypical Brain Maps for Alzheimer's Disease Progression Modeling

**Ahmad Wisnu Mulyadi**[2]                                    WISNUMULYADI@KOREA.AC.KR

**Heung-Il Suk**[1,2,*]                                          HISUK@KOREA.AC.KR

[1] *Department of Artificial Intelligence, Korea University, Seoul, Republic of Korea*

[2] *Department of Brain and Cognitive Engineering, Korea University, Seoul, Republic of Korea*

## Abstract

Discovering the brain progression over a lifetime is beneficial for identifying the subject affected by neurodegenerative disorders, such as Alzheimer's disease (AD) which require detection at the earliest possible time for the sake of delaying the progression by the virtue of particular treatments. As brain progressions in terms of both normal aging and AD-pathology tend to be entangled to each other, distinguishing the progression pathways of AD over the normal aging brains is quite an intricate task. To this end, we propose Prototypical Brain Maps (ProtoBrainMaps) for modeling the AD progressions through the established prototypes in the latent space via clinically-guided topological maps. Having devised as an interpretable network, it possesses the ability to establish and synthesize a set of well-interpolated prototypical brains, each corresponding to certain health conditions in terms of neurodegenerative diseases.

**Keywords:** Alzheimer's Disease, Disease Progression Modeling, Deep Generative Model

## 1. Introduction

Alzheimer's disease (AD) is a well-known illness as an irreversible and progressive neurodegenerative disorder and the most common cause of dementia, compelled the necessity of identification at the earliest possible stage and time. In identifying such a disease, the structural magnetic resonance imaging (sMRI) of the brain plays an essential role as it provides macroscopic visual information and could be further exploited to extract the potential informative biomarkers (*i.e.*, atrophic changes, expanded ventricles, *etc.*) (Apostolova and Thompson, 2008; Suk et al., 2015). However, these morphological changes are quite subtle and bound to entangled between normal aging and neurodegenerative diseases (Sivera et al., 2019), thus recognizing whether or not the subject converting towards AD in the near follow-up years remains a longstanding and challenging task.

In this preliminary study, we take a novel approach in modeling the brain progression in terms of neurodegenerative disease through the integration of deep generative models with clinically-guided clustering networks by utilizing the baseline (first-visit) 3D sMRI. In addition, as a set of neurodegenerative disease measurements (*i.e.*, cognitive score, clinical stage) holds essential facts regarding the underlying subject's health conditions, we also exploit such factors as the auxiliary clinical information. Devised as an interpretable network, the proposed framework possess the ability to establish as well as synthesize a set of representative prototypical brains, each with certain and visually distinct morphological traits, thus, mimicking the progression of neurodegenerative disease.

---

\* Corresponding author

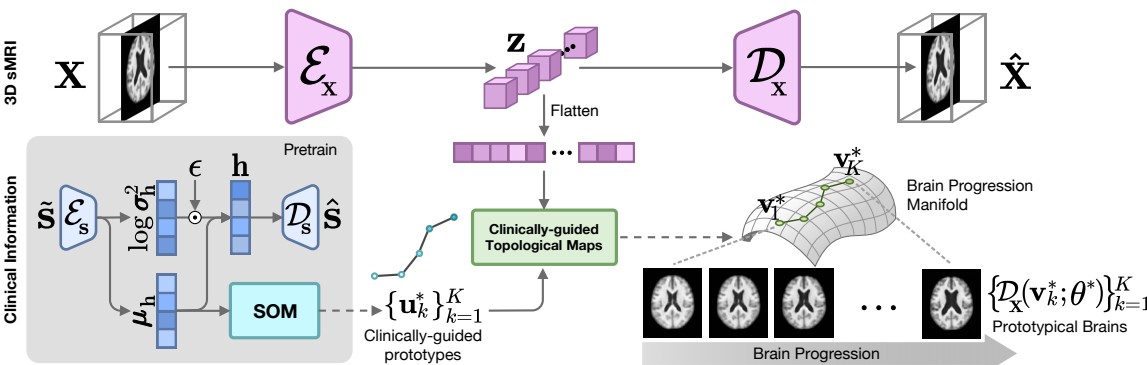

Figure 1: Overall architecture of ProtoBrainMaps.

## 2. Proposed Method

As depicted in Fig. 1, we propose Prototypical Brain Maps (ProtoBrainMaps) as a novel interpretable framework in exploiting AD progression modeling by unifying deep generative models to obtain meaningful latent representations from two modalities, which are convolutional autoencoders (CAEs) for the 3D brain sMRI images, and variational autoencoders (VAEs) for the clinical information (*i.e.*, cognitive score, clinical stage), respectively; followed by a clinically-guided topological maps (CTMs). We devise such CTMs upon a supervise-variant of the self-organizing map (SOM) (Kohonen, 1990) to utilize the clinical factors as auxiliary information to further assist the framework in establishing the faithful brain progression manifold. Thereby, we imbue the interpretability to the network such that it has a way to synthesize a set of representative 3D prototypical brains that reflects the neurodegenerative disease progressions.

## 3. Preliminary Experiments and Discussion

We conducted preliminary experiments on Alzheimer's Disease Neuroimaging Initiative (ADNI) dataset (Mueller et al., 2005). Firstly, we investigated the brain progression manifold established by ProtoBrainMaps using tSNE as depicted in Fig. $2(a)$, $2(c)$. The decoded information from those clinical features is also presented in Fig. $2(b)$, indicating cognitive score and clinical stage progression which were aligned with the neurodegenerative progression. Furthermore, given the established prototypes $\mathbf{V}^*$ in Fig. $2(c)$, we synthesized a set of prototypical brains by means of the trained decoder $\mathcal{D}_{\mathbf{X}}$ through $\{\mathcal{D}_{\mathbf{X}}(\mathbf{v}_k^*; \theta^*)\}_{k=1}^K$. As shown in Fig. $2(d)$, the proposed method accomplished in generating quite well-interpolated prototypical brains with $K = 15$. Eventually, the differences of the generated prototypical brains exhibited apparent morphological changes of brains portrayed by both aging and neurodegenerative diseases' traits. For instance, it revealed the brain growth (volume) changes (highlighted in blue) as well as atrophic changes (highlighted in red) in certain brain areas, which are commonly associated with AD (*i.e.*, temporal lobes, hippocampal, ventricle). These results demonstrated that the proposed method was able to discover the faithful underlying neurodegenerative disease progression manifold. As future works, more extensive experiments are possible to be carried out upon these initial findings, *i.e.*, to better utilize the established prototypical brains for improving the diagnostic downstream tasks.

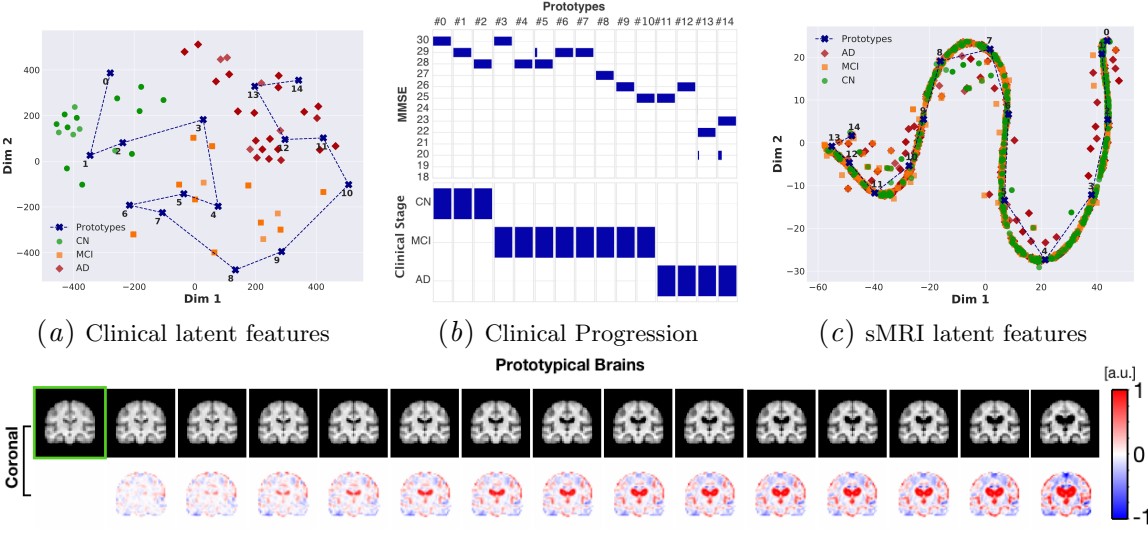

($a$) Clinical latent features     ($b$) Clinical Progression     ($c$) sMRI latent features

($d$) Coronal slice of decoded 3D prototypical brains

Figure 2: ProtoBrainMaps were able to discover: (a) clinical latent features, (b) decoded clinical progression, (c) sMRI latent features, and (d) 3D prototypical brains.

## Acknowledgments

This work was supported by Institute of Information communications Technology Planning & Evaluation (IITP) grant funded by the Korea government (MSIT) (No. 2017-0-01779, A machine learning and statistical inference framework for explainable artificial intelligence, and No. 2019-0-00079, Artificial Intelligence Graduate School Program (Korea University)).

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
