# OpenReview forum: "ProtoBrainMaps: Prototypical Brain Maps for Alzheimer's Disease Progression Modeling"
_MIDL.io/2021/Conference/Short — MIDL 2021 Poster_

### Official Review · Reviewer_ocMH · 2021-05-04

**Confidence:** 4
**Final Rating:** 3

**Summary:**

This paper proposes an interesting approach to modeling Alzheimer's disease progression using *self-organizing maps* on latent features from two different modalities. The authors employ a convolutional autoencoder to map 3D brain sMRI images to a latent representation and combine this with the latent feature of a variational autoencoder trained on auxiliary clinical information. Both latent representations are combined using a supervised version of SOM, which the authors refer to as *clinically-guided topological maps*. The CTMs are used to deduce a brain progression manifold in which $K=15$ prototypical brain interpolations are identified, representing different clinical progression on an ordinal scale. The decoder of the sMRI autoencoder can be used to map the prototypical brains back to the image domain and create visualizations that match the respective AD progression.

**Strengths:**

* The paper is well-written and easy to follow
* Fig. 1 provides a clear overview of the proposed method
* Preliminary experiments show the ability of the method to interpolate between different clinical stages of AD progression
* It proposes a novel approach to an interesting problem and fits well to the scope of MIDL

**Weaknesses:**

* Few details on how the CTM is computed
* AD and age-related feature entanglement is used as motivation, but not addressed by the proposed method
* No code repository is provided yet (though the authors promise to do so upon publication)

**Deanonymize Review:**

no

**Detailed Comments:**

Overall, I find this paper very interesting, even though it does not outline a clinical problem that can be addressed with the proposed approach directly. I think most of its shortcomings can be attributed to the strict 3 page limit. The authors motivate their work by the entanglement of age- and AD-related morphological changes but do not state how their approach can assist or disentangle this (maybe by incorporating the auxiliary information?). Moreover, I would anticipate more details on the implementation of the supervised SOM and how it is used to derive the brain progression manifold. I think the introduction could be shortened in order to accommodate 1–2 more sentences on the CTM.

**Justification Of The Rating:**

The paper is a valuable addition to MIDL 2021 and I would like to get more details about the CTM in a (poster) presentation. However, I think the paper would benefit from shortening the introduction and providing more details about the method.

**Paper Type:**

methodological development

**Special Issue:**

no

---

### Official Review · Reviewer_U4dr · 2021-05-07

**Confidence:** 5
**Final Rating:** 2

**Summary:**

The paper presents a combination of separate convolutional autoencoder for structural MRI and variational autoencoders for the clinical information for the modeling of brain progression in Alzheimers disease. Using a tSNE mapping of the combined/aggregated latent space of the two AEs, they then establish prototypical images representing the progression via simple clustering.

**Strengths:**

- combination of structural MRI and clinical information, each with their own, separate AE
- applied to Alzheimer disease which shows significant longitudinal progression
- using ADNI data
- idea of generating prototypical brain image is interesting

**Weaknesses:**

- quite a number of things/details are unclear about the method,
- it seems that the two AE are not linked, so the latent representation for structural images and clinical data are wholly separated and then finally combined to compute the two-dimensional tSNE mapping.
- each step is rather straightforward and well known (AEs, tSNE mapping, and clustering in tSNE mapping)
- simplistic when compared to the current work in the deep learning field
- this is not longitudinal, but rather a cross-sectional method (only using baseline ADNI data)

**Deanonymize Review:**

no

**Justification Of The Rating:**

- No real novelty here, the approach is rather straightforward and the prototypical images are simple clusters in the t-SNE map (connected along a closest distance path)
- Still some interesting results and thus only a weak reject

**Paper Type:**

methodological development

**Special Issue:**

no

---

### Meta-Review · Area_Chair_eg6C · 2021-05-11

**Recommendation:** Accept (Poster)
**Confidence:** 4

**Metareview:**

The reviewers are split in their opinion, while both find some interest in the application some more clinical motivation could be given. While the second reviewer comments in overall positively on the method the first reviewer points out several weaknesses (not link between AEs, combination of known steps). I agree that the work is borderline but I am confident that by addressing the detailed comments: slightly improving and expanding the method description, make sure to provide the promised link to source code and potentially shorten the lengthy introduction - this can be made into a good MIDL paper.

---

### Decision · Program_Chairs · 2021-05-11

Accept (Poster)